# Induced Pluripotent Stem Cell-Derived Cardiomyocytes with *SCN5A* R1623Q Mutation Associated with Severe Long QT Syndrome in Fetuses and Neonates Recapitulates Pathophysiological Phenotypes

**DOI:** 10.3390/biology10101062

**Published:** 2021-10-18

**Authors:** Emiko Hayama, Yoshiyuki Furutani, Nanako Kawaguchi, Akiko Seki, Yoji Nagashima, Keisuke Okita, Daiji Takeuchi, Rumiko Matsuoka, Kei Inai, Nobuhisa Hagiwara, Toshio Nakanishi

**Affiliations:** 1Department of Pediatric Cardiology and Adult Congenital Cardiology, Tokyo Women’s Medical University, 8-1 Kawada-cho, Shinjuku-ku, Tokyo 162-8666, Japan; yfurutani@twmu.ac.jp (Y.F.); nanako.res@gmail.com (N.K.); takeuchi.daiji@twmu.ac.jp (D.T.); inai.kei@twmu.ac.jp (K.I.); nakanishi.toshio@twmu.ac.jp (T.N.); 2Department of Preventive Medicine, Tokyo Women’s Medical University, 8-1 Kawada-cho, Shinjuku-ku, Tokyo 162-8666, Japan; nishii.akiko@twmu.ac.jp; 3Department of General Medicine, Tokyo Women’s Medical University, 8-1 Kawada-cho, Shinjuku-ku, Tokyo 162-8666, Japan; 4Department of Cardiology, Tokyo Women’s Medical University, 8-1 Kawada-cho, Shinjuku-ku, Tokyo 162-8666, Japan; hagiwara.nobuhisa@twmu.ac.jp; 5Department of Surgical Pathology, Tokyo Women’s Medical University, 8-1 Kawada-cho, Shinjuku-ku, Tokyo 162-8666, Japan; nagashima.yoji@twmu.ac.jp; 6Center for iPS Cell Research and Application (CiRA), Kyoto University, Kyoto 606-8507, Japan; okita@cira.kyoto-u.ac.jp; 7Wakamatsukawada Clinic, 10-7 Kawada-cho, Shinjuku-ku, Tokyo 162-0054, Japan; rumimatsu@gmail.com

**Keywords:** *SCN5A*, Nav1.5, mutation, R1623Q, induced pluripotent stem cells, cardiomyocyte, differentiation, neonatal, congenital long QT, automated patch-clamp

## Abstract

**Simple Summary:**

In this study, the induced pluripotent stem cell-derived cardiomyocyte model from a patient with long QT syndrome harboring a heterozygous Nav1.5 R1623Q mutation exhibited prolonged field potential duration corrected by Fridericia’s formula (FPDcF, analogous to QTcF). FPDcF was shortened with mexiletine treatment and increased the frequency of arrhythmia-like EAD events following E4031, an I_kr_ blocker, administration. These characteristics partly reflect the patient phenotypes. As the R1623Q mutation is related to severe congenital LQT syndrome in fetuses and neonates, the effect of the neonatal variants on the electrophysiological properties of the R1623Q mutant was examined using an automated patch-clamp system. Our results demonstrated that both R1623Q and neonatal R1623Q delayed inactivation of I_Na_ and increased late Na current. We speculated that neonatal Nav1.5 ameliorates QTc prolongation. Developmental switching of neonatal/adult Nav1.5 isoforms might play a role in the mechanisms underlying severe long QT syndrome in fetuses and neonates.

**Abstract:**

The *SCN5A* R1623Q mutation is one of the most common genetic variants associated with severe congenital long QT syndrome 3 (LQT3) in fetal and neonatal patients. To investigate the properties of the R1623Q mutation, we established an induced pluripotent stem cell (iPSC) cardiomyocyte (CM) model from a patient with LQTS harboring a heterozygous R1623Q mutation. The properties and pharmacological responses of iPSC-CMs were characterized using a multi-electrode array system. The biophysical characteristic analysis revealed that R1623Q increased open probability and persistent currents of sodium channel, indicating a gain-of-function mutation. In the pharmacological study, mexiletine shortened FPDcF in R1623Q-iPSC-CMs, which exhibited prolonged field potential duration corrected by Fridericia’s formula (FPDcF, analogous to QTcF). Meanwhile, E4031, a specific inhibitor of human ether-a-go-go-related gene (hERG) channel, significantly increased the frequency of arrhythmia-like early after depolarization (EAD) events. These characteristics partly reflect the patient phenotypes. To further analyze the effect of neonatal isoform, which is predominantly expressed in the fetal period, on the R1623Q mutant properties, we transfected adult form and neonatal isoform *SCN5A* of control and R1623Q mutant SCN5A genes to 293T cells. Whole-cell automated patch-clamp recordings revealed that R1623Q increased persistent Na^+^ currents, indicating a gain-of-function mutation. Our findings demonstrate the utility of LQT3-associated R1623Q mutation-harboring iPSC-CMs for assessing pharmacological responses to therapeutic drugs and improving treatment efficacy. Furthermore, developmental switching of neonatal/adult Nav1.5 isoforms may be involved in the pathological mechanisms underlying severe long QT syndrome in fetuses and neonates.

## 1. Introduction

Type 3 long QT syndrome (LQT3), which is caused by *SCN5A* gain-of-function mutations, is a potentially life-threatening disease with a high risk of sudden cardiac death resulting from torsade de points (TdP) [1,2,3]. *SCN5A* encodes the α subunit of the cardiac voltage-gated sodium channel (Nav1.5), which allows Na^+^ flux in response to a depolarizing stimulus [4]. Nav1.5 comprises four homologous domains (DI–DIV), and each domain contains six transmembrane segments (S1–S6). Several mutations have been reported in *SCN5A*. We were the first to report a de novo missense mutation of *SCN5A* R1623Q at an external position in segment S4 of the highly conserved DIV in a female Japanese patient [5]. DIV is important for the inactivation of sodium channels [6]. Transfection of wild-type (WT) and R1623Q mutant genes into oocytes and HEK293 cells resulted in delayed repolarization [7]. Single-channel analysis revealed that the channels were open for a prolonged duration [8,9]. Miller et al. [10] reported that recurrent third-trimester fetal loss or sudden infant death can result from the R1623Q mutation. Previous studies have reported that the *SCN5A* R1623Q mutation is one of the most frequent genetic variations in severe congenital LQT syndrome among fetuses and neonates [5,10,11,12,13,14,15,16,17,18].

Induced pluripotent stem cells (iPSCs), which are established from human cells [19,20], can differentiate into cardiomyocytes [21]. Hence, various in vitro models of LQT3 have been established using patient iPSC-derived cardiomyocytes (iPSC-CMs), including cells with the deletion of lysine-proline-glutamine (ΔKPG) in the intracellular loop between DIII and DIV [22], V240M mutation in DI [23], R535Q mutation in DII [23], and R1644H [24] and N1774 mutations [25] in DIV. However, iPSC-CMs from patients with LQT3 harboring R1623Q mutations have not yet been established. In this study, iPSC-CMs were established from healthy individuals and a male patient with R1623Q mutation diagnosed with severe LQT3 at the age of 1 month. The pharmacological properties, including responses to the Na channel blocker mexiletine, of the established iPSC-CMs were examined using a multi-electrode array system.

The major alternative splicing variant of *SCN5A* involves the inclusion of two alternative exons (5′-exon (6a variant, ‘neonatal’), and 3′-exon (6 variant), ‘adult’) encoding the DI segment S3 and D1 S3/S4 extracellular linker [26]. The mammalian Na channel exon 6a variant was first described in the rat brain Nav1.2 and Nav1.3 [27,28] and is predominantly expressed in neonates but is gradually replaced with the adult type. Consequently, transcripts harboring 5′-exon (6a) and 3′-exon are termed ‘neonatal’ and ‘adult,’ respectively. Chioni et al. [29] demonstrated that neonatal Nav1.5 is expressed in several tissues, including the brain and heart, during development, and that the expression in neonates was significantly higher than that in adults. This ‘neonatal’ isoform differs from the ‘adult’ isoform at seven amino acids in the extracellular loop between S3 and S4 of DI. Murphy et al. [30] also demonstrated that the expression of neonatal Nav1.5 in the heart of the fetus is higher than that in the heart of infants or adults. The neonatal Nav1.5 isoform with L409P mutation and R558 polymorphism exhibited a more pronounced shift in fast inactivation and larger persistent Na^+^ current than the adult isoform [30]. Walzik S et al. [31] also reported that the neonatal variant affected the Nav1.5 channel kinetics. On the basis of these investigations, we hypothesized that the neonatal isoform of Nav1.5 may affect the electrophysiological properties of the R1623Q mutant.

In this study, two experiments were performed to investigate the properties of the Nav1.5 R1623Q mutant. A patient-derived iPSC model harboring a heterozygous *SCN5A* R1623Q mutation was established, and its properties and pharmacological responses were characterized. As the R1623Q mutation is related to severe congenital LQT syndrome in fetuses and neonates, the effect of the neonatal isoform on the electrophysiological properties of the R1623Q mutant was examined using stable expression 293T cell lines with a recently developed automated patch-clamp recording technology.

## 2. Materials and Methods

### 2.1. Patients

This study was reviewed and approved by the central ethics board at Tokyo Women’s Medical University (approval number: 2910-R5) to ensure that the procedures were performed according to the guidelines of the Declaration of Helsinki and the ethical standards of the responsible committee on human experimentation. Written informed consent was obtained from all individuals enrolled in the study before genetic and clinical investigations according to the guidelines of the Declaration of Helsinki and the university’s ethics committees. As the proband could not provide consent to decisions about participation in research, consent was obtained from the parents.

### 2.2. Generation of iPSCs from Lymphoblastoid Cell Lines (LCLs)

To generate LCLs, we infected the peripheral blood lymphocytes with the Epstein–Barr virus. LCLs were cultured in GIT medium (Kohjin Bio Co., Ltd., Saitama, Japan) at 37 °C and 5% CO_2_ in a humidified incubator. Propagated LCLs were cryopreserved in liquid nitrogen until analysis. The LCLs were reprogrammed into iPSCs through electroporation of episomal plasmids encoding the reprogramming genes OCT3/4, SOX2, KLF4, L-MYC, LIN28, and mouse p53DD (c-terminal dominant-negative) using a NEPA21 electroporator (Nepa Gene Co., Ltd., Chiba, Japan) [32]. After culturing the LCLs in StemFit AK02N medium (Ajinomoto Healthy Supply Co., Inc., Tokyo, Japan) under feeder-free conditions for 3–4 weeks, the visible iPSC-like colonies were picked and cultured on iMatrix-511 (laminin-511 E8, Nippi. Inc., Tokyo, Japan)-coated wells. The medium was replaced three times a week. In the present study, we established six iPSC clones from R1623Q patient-derived B cells and multiple clones from five healthy controls. Given individual, clonal, and batch differences, we used the best available clones and batches for cardiomyogenesis and repeated the differentiation experiments.

### 2.3. Characterization of Reprogrammed Cells

#### 2.3.1. Immunocytochemistry

Colonies of iPSCs and iPSC-CMs were fixed with 4% paraformaldehyde (PFA)/ phosphate-buffered saline (PBS) (FUJIFILM Wako, Osaka, Japan) for 20 min, washed twice with PBS (FUJIFILM Wako), permeabilized with 0.2% Triton X-100 in PBS for 5 min, and blocked with 3% bovine serum albumin (BSA) in Tris-buffered saline containing 0.1% Tween-20 (TBST). The fixed iPSCs were incubated with anti-TRA-1-60 (1:150; GTX48033, GeneTex, LA, USA), anti-SSEA4 (1:150; ab16287, Abcam, Cambridge, UK), and anti-OCT4 (1:150; ab19857, Abcam) antibodies in 1% BSA/TBST for 1 h at room temperature or overnight at 4 °C. Meanwhile, the fixed iPSC-CMs were incubated with anti-α-actinin (1:150; A7732, Sigma-Aldrich, St. Louis, MO, USA) antibodies. The cells were washed thrice with TBST and incubated with rhodamine–phalloidin, 4′-6-diamidino-2-phenylindole (DAPI), and the following secondary antibodies in 1% BSA/TBST for 1 h at room temperature or overnight at 4 °C: Alexa 488-conjugated anti-mouse IgG (1:150; Thermo Fisher Scientific, Waltham, MA, USA), Alexa 488-conjugated anti-mouse IgM (1:150; Thermo Fisher Scientific), or Alexa 488-conjugated anti-rabbit IgG (1:150; Thermo Fisher Scientific). Next, the cells were washed thrice with TBST and analyzed under a BZ-X700 fluorescence microscope (Keyence Corporation, Osaka, Japan).

#### 2.3.2. Sequence and Karyotype Analyses

The R1623Q mutation in patient peripheral lymphocytes or patient-derived iPSCs (Appendix A) was confirmed using DNA sequencing. The region encoding the mutation in the patient iPSC genome was amplified using polymerase chain reaction (PCR). The PCR products were purified and sequenced using the original primers (Appendix A).

Karyotyping of iPSCs was performed at the Nihon Gene Research Laboratories Inc. (Miyagi, Japan) following standard protocols for high-resolution G-banding.

#### 2.3.3. Teratoma Formation

All experiments were conducted according to a protocol approved by the Institutional Animal Experiment Committee of the Tokyo Women’s Medical University (AE16-119, AE17-70, AE18-128-B). All applicable international, national, and/or institutional guidelines for the care and use of animals were followed. All procedures performed in studies involving animals were in accordance with the ethical standards of the institution. Approximately 1 × 10^6^ iPSCs were inoculated into the renal subcapsular regions of SCID mice (CB-17/Icr-scid/scid) to form teratomas. The in vivo differentiation capacity of iPSCs was examined. Teratomas were harvested and fixed in 4% PFA/PBS for histological analysis after 8 weeks. The paraffin-embedded sections were prepared following a standard protocol and stained with hematoxylin and eosin. The sections were examined and imaged under a microscope.

### 2.4. Differentiation of Cardiomyocytes from iPSCs

Day 0: The iPSC colonies were isolated by treatment with PBS containing ethylenediaminetetracetate (EDTA; 0.5 mM), and single cells were obtained. The cells (3.5–4.0 × 10^5^ cells/cm^2^) were seeded onto ultra-low attachment multiple 6-well plates (Corning Inc., New York, NY, USA) or 90 mm culture dishes (PrimerSurface, MS-9090, Sumitomo Bakelite, Co., Ltd., Tokyo, Japan) and cultured in complete StemPro 34 serum-free medium (SFM) containing supplement (StemPro-Nutrient Supplement), 1× monothioglycerol (FUJIFILM Wako), 50 μg/mL ascorbic acid (FUJIFILM Wako), 100 U/mL penicillin G/100 U/mL streptomycin (FUJIFILM Wako), and 10 μg/mL Y-27632 (FOCUS Biomolecules, Plymouth Meeting, PA, USA) at 37 °C and 5% CO_2_.

Day 1 (mesodermalization): Cell aggregates (embryoid bodies, EBs) were formed from dissociated iPSCs. The cultures including the EBs were supplemented with an equal volume of complete StemPro34 SFM without Y-27632. Bone morphogenetic protein-4 (BMP4; R&D Systems Inc., Minneapolis, MN, USA), activin A (Nacalai Tesque Inc., Kyoto, Japan), and basic fibroblast growth factor (bFGF; ReproCELL Inc., Kanagawa, Japan) were added to the medium to final concentrations of 2.5, 1.25, and 5 ng/mL, respectively.

Day 3 (cardiomyogenesis): Most of the medium was removed, and the cells were incubated with complete StemPro34 SFM containing 10 μM KY02111 (MedChemExpress, Monmouth Junction, NJ, USA), 10 μM XAV939 (Abcam), and 10 ng/mL human vascular endothelial growth factor (hVEGF; Thermo Fisher Scientific) without Y-27632.

Days 7 and 10: Half the volume of the medium was replaced with complete StemPro34 SFM containing 10 μM KY02111, 10 μM XAV939, and 10 ng/mL hVEGF without Y-27632.

Days 8 to 10: Spontaneously beating cardiomyocyte EBs were obtained.

Day 15: The spontaneously beating cardiomyocyte aggregates were resuspended in 1× TrypLE Express (Thermo Fisher Scientific) and seeded into the culture-treated multiple 6-well plates with Dulbecco’s modified Eagle’s medium/Ham’s Nutrient Mixture F-12 (DMEM/F-12, FUJIFILM Wako) containing 10% fetal bovine serum (FBS, Biowest, Nuaille, France), 1× GlutaMAX (Thermo Fisher Scientific), 1× minimal essential medium (MEM) non-essential amino acids (NEAAs) (FUJIFILM Wako), and 100 U/mL penicillin G/100 U/mL streptomycin. The medium was replaced with fresh medium every two to three days.

Days 20 to 22: The cells were cultured in DMEM/F-12 supplemented with 4 mM lactate (FUJIFILM Wako), 3.7 g/L sodium hydrogen carbonate (FUJIFILM Wako), and 0.04% phenol red (FUJIFILM Wako) for three days to specifically select cardiomyocytes, following the protocols of Tohyama et al. [33,34]. Next, the medium was replaced with DMEM/F-12 containing 10% FBS, 1× GlutaMAX, 1× MEM NEAAs, and 100 U/mL penicillin G/100 U/mL streptomycin. The medium was replaced with fresh medium once every two to three days. The cardiomyocytes were further purified using a PSC-derived cardiomyocyte isolation kit and magnetic-activated cell sorter (Miltenyi Biotec B.V. & Co., Bergisch Gladbach, Germany).

Days 27 to 30: The purified iPSC-CMs were detached with 0.5× TrypLE Express. The cells were then cultured on vitronectin (50 μg/mL; 3 to 3.5 μL/probe; Sigma-Aldrich)-coated MED64 probes (R515A, Alpha MED Scientific Inc., Osaka, Japan) at a density of 2 to 3 × 10^4^ cells in 2 to 3 µL of the culture medium described above at 37 °C and 5% CO_2_ for 2 h. Next, 2 mL of the culture medium was added to each probe. The medium was changed every three to four days. The culture was maintained on the probes for 7 to 10 days to obtain a sheet of cardiomyocytes with spontaneous and synchronous electrical activity.

### 2.5. Characterization of iPSC-CMs

#### 2.5.1. Quantitative Real-Time PCR (qRT-PCR)

Total RNA was isolated using the FastGene RNA Kit (Nippon Genetics Co., Ltd., Tokyo, Japan) and reverse-transcribed to complementary DNA (cDNA) using PrimeScript RT master mix (Takara Biotechnology Co., Ltd., Shiga, Japan), following the manufacturer’s instructions. Human normal adult heart atrium (female, 57 years of age) and human fetal heart (male, 31-week-old) total RNAs (BioChain Institute Inc., Newark, CA, USA) were used as control. The expression levels of target genes were examined using qRT-PCR with Thunderbird SYBR qPCR (Toyobo, Co. Ltd., Osaka, Japan) and a Pico Real PCR System (Thermo Fisher Scientific). The PCR conditions were as follows: initial denaturation at 95 °C for 60 s, followed by 40 cycles of 95 °C for 15 s (denaturation) and 60 °C for 45 s (annealing). The primers used in qRT-PCR analysis are shown in Appendix A. The housekeeping gene glyceraldehyde 3-phosphate dehydrogenase (GAPDH) was used as an internal control. The expression of the target mRNA relative to that of GAPDH in samples was calculated using the 2 ^−ΔΔCT^ method (where CT is the threshold cycle, ΔCT = CT_gene of interest_ − CT_GAPDH_, and ΔΔCT = ΔCT_sample_ − ΔCT_fetal heart_). Each sample was measured in duplicate as technical replicates.

#### 2.5.2. Field Potential Recording Using the MED64 System and Data Analysis

The MED64 probes with iPSC-CM sheets were transferred to the MED connector (MED-C03, Alpha MED Scientific) and equilibrated under a MED connector cover (MED-CC06, Alpha MED Scientific) filled with humidified 5% CO_2_ in 2 mL of fresh medium. The field potential was recorded for approximately 30 min (baseline) before drug treatment. Baseline recordings were used to identify the best electrode to obtain further recordings. For each drug concentration, field potential was recorded for at least 15 min and the recordings from the last 1 min were extracted as a dataset for analysis of field potential duration (FPD), inter-spike interval (ISI), and waveforms. All measurements using the MED64 system were performed at 37 °C. Beats per minute were calculated from the ISI (s) in 1 min. The criterion of accepted data from MED64 measurement was based on baseline field potential waveforms. Waveforms in which the second peak amplitude was >100 μV, and the ISI was <2.5 s, were employed for further analysis. We repeatedly utilized iPSC-CM probes, at intervals of few days, until day 75. The FPD values were corrected for the beating rate (corrected FPD: FPDc) using Fridericia’s formula as follows: QTcF = QT/(RR interval)^1/3^ (where QT and RR indicate FPD and ISI, respectively). The absolute FPDcF values at each drug concentration were normalized as a percentage change from the baseline values. The stock solutions of E4031 (10 mM; FUJIFILM Wako), isoproterenol hydrochloride (50 mM; Sigma-Aldrich), and mexiletine hydrochloride (100 mM; FUJIFILM Wako) were prepared in Milli-Q water and diluted to the required concentrations in the culture medium of the probe.

### 2.6. Construction of Expression Plasmids

R1623Q site-directed mutagenesis: The WT human *SCN5A* cDNA cloned into pET19b from the original pcDNA construct hH1 (a kind gift from Dr. Roland G. Kallen, University of Pennsylvania, School of Medicine, Philadelphia, PA, USA) was used as a template to engineer the mutation R1623Q using the QuickChange II XL-site-directed mutagenesis kit (Agilent Technologies, Santa Clara, CA, USA) and ‘mutagenesis primers’ (Appendix A). The resultant construct was directly sequenced to verify the presence of the desired mutation and the absence of additional variations. The mutated *SCN5A* coding region cDNAs were subcloned into the mammalian expression vector pSF-CMV-Ub-hygro-SV40 ORI SBFI (Sigma-Aldrich) to generate the stable 293T cells. The 293T cells were isolated from human embryonic kidney (HEK) 293 cells and were transformed using large T antigen. The pSF-CMV-Ub-hygro-SV40 ORI SBFI-SCN5A constructs included a FLAG tag (DYKDDDDK) at the N-terminus of SCN5A.

*SCN5A* neonatal variant: The *SCN5A* cDNA containing the exon 6a region was cloned from cDNA fragments, which were obtained after reverse transcription of fetal heart total RNA (BioChain Institute) using ‘cloning of *SCN5A* exon 6a region primers’ (Appendix A). The fragments were cloned into the XhoI sites of the *SCN5A* coding region using in-fusion cloning (Takara Biotechnology). The coding region cDNAs of the neonatal variant *SCN5A* were subcloned into pSF-CMV-Ub-hygro-SV40 ORI SBFI as described above.

### 2.7. Cell Culture and Generation of Nav1.5 Stable Expression Cell Lines

The 293T cells were cultured in high-glucose DMEM (FUJIFILM Wako) containing 10% FBS (Biowest) and 100 U/mL penicillin G/100 U/mL streptomycin at 5% CO_2_ and 37 °C. To subject the cells expressing the channels to automated patch-clamp recordings, we screened the 293T cell lines stably expressing Nav1.5 proteins. Transient transfections were performed using the Lipofectamine 2000 reagent (Thermo Fisher Scientific). The 293T cells were seeded in 12-well plates and transfected with 0.8 μg plasmid (plasmid (μg): Lipofectamine 2000 (μL) = 1:2.5) for two to three days. The cells were digested with 0.5× TrypLE Express and the cultures were transferred to 100 mm culture dishes. Next, the cells were cultured in DMEM containing 100 μg/mL hygromycin, 10 μg/mL G418, and 10% FBS. The resistant cell colonies were picked using aseptic tips with a pipette and transferred to the wells of a 96-well plate for further culture. Copies of the culture in the 96-well plates were made to examine the expression of FLAG-tagged Nav1.5 using immunocytochemistry when the cells reached 90% confluency. The 293T cells were fixed and permeabilized using methanol for 20 min and blocked with 3% BSA/TBST. The samples were then incubated with anti-FLAG M2 monoclonal antibodies (Sigma-Aldrich), followed by incubation with Cy3-conjugated anti-mouse secondary antibodies (Sigma-Aldrich). The images were captured using a BZ-X700 fluorescence microscope (Keyence). The cells expressing the channels were transferred to duplicate 12-well plates for further culture with DMEM containing 100 μg/mL hygromycin, 10 μg/mL G418, and 10% FBS. After the confluency of the cells in the 12-well plates reached approximately 90%, the expression of Nav1.5 proteins in one of the duplicate wells was examined using Western blotting. The cells subjected to these selection steps were allowed to proliferate and were subjected to automated patch-clamp recordings.

### 2.8. Automated Patch-Clamp Recording

Sodium currents were measured at room temperature in a whole-cell configuration using a Syncropatch 384 PE (Nanion Technologies). Pulse generation and data collection were performed using PatchController 384 V.1.3.0 and DataController 384 V1.2.1 (Nanion Technologies, München, Germany). Whole-cell currents were filtered at 3 kHz and acquired at 10 kHz. Access resistance and apparent membrane capacitance were estimated using the built-in protocols. Human embryonic kidney 293T cells stably expressing WT or mutant Nav1.5 were plated into collagen-coated 100 mm culture dishes 48 to 72 h before the experiment. On the day of the experiment, the cells (60% to 80% confluency) were washed once with Hank’s balanced salt solution (HBSS, no calcium, no magnesium, no phenol red, Thermo Fisher Scientific), detached with cold 1× TrypLE Express, and resuspended in OptiMEM (Thermo Fisher Scientific) and HBSS. The cell suspension was added to each well of a 384-well single hole “chip” (Nanion Technologies). Whole-cell currents were recorded at room temperature in a whole-cell configuration. The holding potential for all experiments (−100 mV) and specific voltage-clamp protocols are depicted as figure insets. The composition of the external solution was as follows: 3.5 mM triethanolamine (TEA), 135 mM NaCl, 4 mM KCl, 3.5 mM CaCl_2_, 10 mM HEPES, and 10 mM glucose (pH 7.4). Meanwhile, the composition of the internal solution was as follows: 110 mM CsF, 10 mM CsCl, 10 mM NaCl, 10 mM HEPES, and 10 mM ethylene glycol-bis(β-aminoethyl ether)-N,N,N′,N′-tetra acetic acid (EGTA) (pH 7.2). The peak current was normalized for cell capacitance and plotted against voltage to generate the peak current density–voltage curve. A Boltzmann function (I = I/[1 + exp(Vt − V_1/2_)/k)], where k is the slope factor) was fitted to the availability and activation curves to determine the membrane potential eliciting half-maximal activation/inactivation (V_1/2_). The time constant (τ), τ_slow_, and τ_fast_ values were obtained by fitting the current elicited with the macroscopic sodium current protocol to a second-order exponential function.

### 2.9. Statistical Analysis

Statistical analysis of MED64 data was performed using a two-tailed Student’s *t*-test in Excel (Microsoft, Redmond, WA, USA). The results of patch-clamp experiments were analyzed using Clampfit 10.7 (Molecular device LLC., San Jose, CA, USA), two-way analysis of variance, and Scheffe post-hoc test using OriginPro 2019b (LightStone Corp., Tokyo, Japan). Differences were considered significant at *p* < 0.05. Data are presented as mean ± standard error of the mean unless otherwise noted.

## 3. Results

### 3.1. Patient Information and Genetic Analysis

The male proband (Figure 1a) was diagnosed with severe LQT3 (corrected QTc 580 msec) at the age of 1 month (electrocardiogram (ECG) on ECG monitor at the age of 1.5 months is available in Appendix A). A 12-lead ECG recording at age of 2 years showed notched T-waves and torsade de pointes (Figure 1b). Fetal ventricular tachycardia (VT) was documented for the first time at week 29 and day 0. Additionally, a fetal echocardiogram revealed cardiomegaly, pleural effusion, and ascites. Pleural effusion and ascites gradually decreased after daily oral administration of 300 mg flecainide to the mother. As the mother exhibited liver dysfunctions at week 29 and day 5, the dose of flecainide was reduced to 200 mg, which could not prevent VT. The co-administration of lidocaine was effective in suppressing sustained VT. The baby was delivered through cesarean section at week 33 and day 1. Twenty minutes after birth, polymorphic premature ventricular contraction (PVC) and sustained VT were observed. The co-administration of propranolol and mexiletine was effective in controlling sustained VT. At the age of 2 years, the patient developed a VT storm that caused severe brain damage. The patient underwent implantable cardioverter-defibrillator implantation.

Genetic testing revealed a heterozygous *SCN5A* missense mutation c.4868G > A, which resulted in an arginine to glycine substitution at position 1623 of segment 4 in DIV of Nav1.5 (Figure 1c). This arginine1623 is highly conserved from *Drosophila* to human (Figure 1d). However, analysis of parental DNA identified no evidence of germline mosaicism, suggesting that R1623Q occurred de novo in the fetus. No mutations were identified in six other LQT-associated genes (KCNQ1, KCNH2, KCNE1, KCNE2, CACNA1C, and HCN4). Membrane-spanning model of the Nav1.5 channel indicating the locations of exon 6a and R1623Q mutation is shown in Figure 1e.

### 3.2. Generation and Characterization of iPSCs from a Patient Harboring R1623Q Mutation in Nav1.5

iPSCs were generated from LCLs of a patient with Nav1.5 R1623Q mutation (Figure 2a) and healthy volunteers. The morphologies of patient-derived and healthy volunteer-derived iPSCs were not markedly different (Figure 2a and Appendix A). The expression levels of the pluripotent markers TRA-1-60, SSEA4, and OCT4 were confirmed using immunocytochemistry (Figure 2b). The established patient-derived iPSCs exhibited a normal karyotype (Figure 2c) and retained the original R1623Q mutation (Appendix A). Histological analysis of the teratomas revealed the following three germ layers in the tissues: primitive gut-like epithelium (endoderm), cartilaginous tissue (mesoderm), and neuroblasts (ectoderm) (Figure 2d).

### 3.3. Differentiation of R1623Q Mutation-Harboring iPSCs into Cardiomyocytes

The control and R1623Q mutation-harboring iPSCs were differentiated into spontaneously contracting cardiomyocytes from aggregated iPSCs (embryoid body, EB) in suspension culture described in detail in the Materials and Methods section. The schedule of cardiomyocyte differentiation is summarized in Figure 3a. Spontaneously beating EBs appeared on day 8 of differentiation (see Video S1, movie for beating EB on day 15 of differentiation). Sequential analysis of the expressed genes was performed using qRT-PCR. At the initiation of differentiation, the expression of the pluripotency marker OCT4 was downregulated (Figure 3b). The expression of the cardiac marker troponin T (TNNT2) emerged at the start of EB contraction and increased after incubation with lactate-supplemented medium (Figure 3c). Immunostaining analysis revealed the presence of striated muscular structure in R1623Q-iPSC-CMs on day 49 of differentiation (Figure 3d).

### 3.4. MED64 Analysis

The MED64 multi-electrode array system was used to characterize the electrophysiological properties, including FPD, beat rate, and incidence of arrhythmia-like waveforms (early afterdepolarization (EAD) and triggered activity (TA)), of the patient-derived iPSC-CMs. FPD was defined as the interval from the deflection to the peak of the dome and was analogous to the QT interval in a surface ECG. Both WT and R1623Q mutation-harboring iPSC-CMs spontaneously contracted on MED64 probes for more than one month. Representative recordings from WT and R1623Q mutation-harboring iPSC-CMs are shown in Figure 4a. Average FPD was corrected using Fridericia’s formula as described in the Materials and Methods section. FPDcF was significantly prolonged in R1623Q mutation-harboring iPSC-CMs (Figure 4b; **** *p* < 0.0001).

Mexiletine, a Na channel blocker that was used to treat the proband, is commonly prescribed for patients with LQT3. In this study, mexiletine reduced the Na channel current, exhibited as a suppression of the initial upstroke and slope decay (total spike amplitude) (Figure 4c; * *p* < 0.05, ** *p* < 0.01, baseline vs. each dose; no significant differences in percentage changes between WT and R1623Q at each dose level), and dose-dependently increased the beat rate of WT (by up to 118% from the baseline) but decreased the beat rate and FPDcF of R1623Q mutant-harboring iPSC-CMs (by up to 63% and 95%, respectively, from the baseline) (Figure 4d; * *p* < 0.05, ** *p* < 0.01, *** *p* < 0.001, baseline vs. each dose; ★ *p* < 0.05, ★★ *p* < 0.01 WT vs. R1623Q). These data suggest that mexiletine preferentially suppresses QT prolongation in R1623Q-iPSC-CMs but not in WT-iPSC-CMs.

Next, we investigated effect of I_Kr_-blocking drug E-4031. At concentrations between 5 and 100 nM, E-4031 accelerated the beat rate (by up to 128% for WT and 120% in R1623Q-iPSC-CMs, respectively, from the baseline) and prolonged FPDcF of WT by up to 122% from baseline but reduced FPDcF of R1623Q by up to 85% from the baseline (Figure 4e; * *p* < 0.05, ** *p* < 0.01, *** *p* < 0.001, baseline vs. each dose; ★ *p* < 0.05, ★★ *p* < 0.01 WT vs. R1623Q). Additionally, E-4031 promoted the formation of an EAD- and TA-like field potential waveform in the R1623Q mutation-harboring iPSC-CMs but not in the WT iPSC-CMs (Figure 4f,g).

### 3.5. Automated Patch-Clamp Analysis of Neonatal R1623Q Nav1.5 Variant

To examine the effect of neonatal Nav1.5 splice variant on R1623Q mutant currents, we generated five ‘adult’ (= exon 6 spliced variant), four ‘neonatal’ (= exon 6a spliced variant), four ‘R1623Q’ (= exon 6 spliced variant with R1623Q mutation), and four ‘neonatal-R1623Q’ (= exon 6a spliced variant with R1623Q mutation) Nav1.5 stable expression 293T cell lines. I_Na_ with voltage-clamp was analyzed using the automated patch-clamp system. A typical image of Na channel currents with the automated patch clamp system is shown in Appendix A. Representative I_Na_ traces from a folding potential of −100 mV are shown in Figure 5a. The amplitude of peak of the ‘neonatal,’ ‘R1623Q,’ and ‘neonatal-R1623Q’ Nav1.5 was lower than that of the ‘adult’ variant. The fast inactivation of normalized Na^+^ currents at a test potential of −30 mV of both ‘R1623Q’ and ‘neonatal-R1623Q’ Nav1.5 was delayed when compared with that of ‘adult’ and ‘neonatal’ Nav1.5 (Figure 5b). Figure 5c shows the current–voltage relationship of peak I_Na_ (pA/pF) of the ‘adult,’ ‘neonatal,’ ‘R1623Q,’ and ‘neonatal-R1623Q’ Nav1.5. The peak current densities of ‘neonatal,’ ‘R1623Q,’ and ‘neonatal-R1623Q’ Nav1.5 decreased by approximately 44%, 63%, and 55%, respectively, when compared with those of ‘adult’ Nav1.5 (‘adult’: −315.0 ± 44.9 pA/pF at −40 mV; ‘neonatal’: −139.5 ± 11.0 pA/pF at −40 mV; ‘R1623Q’: −199.3 ± 17.3 pA/pF at −40 mV; ’neonatal-R1623Q’: −172.6 ± 9.9 pA/pF at −30 mV). The voltage dependence of the activation of the ‘neonatal’ Nav1.5 shifted toward hyperpolarized potential by 2.5 mV when compared with that of the ‘adult’ Nav1.5. The ‘neonatal-R1623Q’ Nav1.5 shifted toward depolarized potentials by 10.4 mV. The V_1/2_ values of the ‘adult,’ ‘neonatal,’ ‘R1623Q,’ and ‘neonatal-R1623Q’ Nav1.5 were −55.4, −57.9, −52.6, and −45.0 mV, respectively (Figure 5d, Table 1). The voltage dependence of inactivation of the ‘neonatal’ Nav1.5 shifted slightly toward the right when compared with that of the ‘adult’ Nav1.5. The V_1/2_ values in the inactivation curves of the ‘adult,’ ‘neonatal,’ ‘R1623Q,’ and ‘neonatal-R1623Q’ Nav1.5 were −70.1, −68.0, −70.7, and −70.3 mV, respectively (Figure 5e, Table 1). The inactivation τ_slow_ values of ‘R1623Q’ and ‘neonatal-R1623Q’ Nav1.5 were significantly higher than those of ‘adult’ and ‘neonatal’ Nav1.5 at most test pulse voltages between −40 and 40 mV, respectively (Figure 5f). To compare the late I_Na_ of these four Na channels, we normalized the currents to the peak I_Na_. The late I_Na_ of both ‘R1623Q’ and ‘neonatal-R1623Q’ Nav1.5 was higher than that of both ‘adult’ and ‘neonatal’ Nav1.5 (Figure 5g).

## 4. Discussion

In this study, iPSCs from patients with LQT3 harboring a mutant *SCN5A* (R1623Q) and healthy volunteers were established. Spontaneously beating cardiomyocytes were obtained from these iPSCs, which enabled the establishment of the LQT3 disease and control models for further analysis. Prolonged FPDcF, which is analogous to the QT interval on ECG, was observed in R1623Q mutation-harboring iPSC-CMs (Figure 4b). This demonstrated that the LQT phenotype in the developed model was consistent with that reported in previous studies using iPSC-CMs from patients with LQT3 [9]. Pharmaco-electrophysiological analysis revealed isoproterenol dose-dependently increased the beat rate and decreased FPDcF in R1623Q mutant-harboring iPSC-CMs (Appendix A). These results suggested that beta-adrenergic receptors and contraction systems in the developed iPSC-CM model are intact. Mexiletine, a class 1B Na channel blocker that was used to manage the disease in the proband, dose-dependently shortened prolonged FPDcF in R1623Q mutation-harboring iPSC-CMs but not in control iPSC-CMs (Figure 4c). The in vitro model revealed the effectiveness of mexiletine administration in patients with LQT3 harboring the R1623Q mutation. Since no arrhythmia-like events (EAD) occurred in baseline condition of R1623Q mutation-harboring iPSC-CMs, we investigated superimposition effect of E-4031 that inhibits I_Kr_ and prolongs repolarization [35,36,37]. Interestingly, pharmaco-electrophysiological analysis revealed that E-4031 promoted QT prolongation in healthy iPSC-CMs but did not prolong QT in R1623Q mutation-harboring iPSC-CMs (Figure 4d). The persistent Na current prolongs action potential duration and might cancel the effect of inhibition of HERG current. The frequency of EAD in E4031-treated patient-derived iPSC-CMs was higher than that in control iPSC-CMs (Figure 4f,g), consistent with the fact that an increase in the late I_Na_ prolongs APD and leads to reactivation of L-type Ca^2+^ (I_ca,L_) and I_Ca,L_ window currents. This causes depolarization of the membrane to generate EAD [38]. An increase in the dispersion of repolarization provides a potential substrate for reentry. The reagents that enhance the late I_Na_ increase the incidence of EAD and lead to reentrant arrhythmia in the canine, rabbit, and guinea pig hearts [39]. Both adult and neonatal R1623Q mutations enhance the late I_Na_ and can lead to reentrant ventricular tachycardia and TdP in the patient’s heart. These pharmaco-electrophysiological studies using functional iPSC-CMs will provide useful information about not only efficacy of the drugs, but also the toxicity and side effects of drugs on human hearts.

R1623Q is a common mutation in severe fetal and neonatal LQT3 cases. We hypothesized that the neonatal isoform of Nav1.5 may affect the electrophysiological properties of the R1623Q mutant. Initially, we postulated that iPSC-CMs were suitable to investigate differences between neonatal and adult Nav1.5 isoforms. The fraction of adult isoforms gradually increased when compared with those of neonatal isoforms, owing to the improved maturity of iPSC-CM, as previously reported by Veerman et al. [40]. However, given heterogeneity of iPSC-CMs, such as individual, clonal, maturational, and batch differences, we anticipated potential challenges in obtaining conclusive results regarding the contribution of each isoform. Therefore, we performed electrophysiological experiments using 293T cell lines, expressing adult/neonatal Nav1.5 and WT/R1623Q mutant Nav1.5 independently. To examine the effect of neonatal Nav1.5 splice variant on R1623Q mutant currents, we generated stable 293T cells expressing the ‘adult,’ ‘neonatal,’ ‘R1623Q,’ and ‘neonatal-R1623Q’ Nav1.5. These cell lines are suitable for electrophysiological analysis in the automated patch-clamp system, as no additional selection is required before measurement. The peak Na current density of the ‘neonatal-R1623Q’ decreased by approximately 45% and 13% compared with that of the ‘adult’ and ‘R1623Q,’ respectively. The steady-state activation of the ‘neonatal-R1623Q’ Nav1.5 shifted towards the depolarized potentials by 10.4 mV compared with that of the ‘adult’ Nav1.5. These observations suggest a loss-of-function effect by the neonatal isoform. Previous studies [26,31] have reported that the peak current density of the ‘neonatal’ Nav1.5 decreased compared with that of the ‘adult’ Nav1.5, which was consistent with our results. In this study, the ‘R1623Q’ and ‘neonatal-R1623Q’ exhibited significantly increased inactivation τslow values at the maximum test pulse voltages and enhanced the persistent currents, indicating that the R1623Q mutation is a gain-of-function mutation. Kambouris [7] and Makita [9] reported that R1623Q increases the probability of long openings and promotes early reopening, which prolongs the Na current decay. Our results were consistent with those of previous studies [7,9]. However, in this study, we did not observe synergistic effects by the ‘neonatal’ and ‘R1623Q’ on the inactivation τ and late Na currents, which prolonged APD and led to reentrant ventricular tachycardia and TdP.

Murphy et al. [30] reported that compared with those of ‘adult’ Nav1.5, the levels of the neonatal Nav1.5 mRNA were approximately 1.5-fold higher in the fetal heart (28–33 weeks gestation), approximately equal in the infant heart (2–6 months of age), and low (10–20%) in the adult heart (>18 years of age). The authors concluded that SCN5A exhibits a developmental switch in exon 6/6a alternative splicing in the human heart during the early postnatal period. We also compared the levels of the neonatal Nav1.5 mRNA with those of the ‘adult’ Nav1.5, which were 71.7% in fetal heart (male, 31 weeks gestation) and 3.8% in the adult left ventricle (male, 21 years of age) (Appendix A). This finding is consistent with the conclusion of Murphy et al. [30] and suggests existence of individual differences. Walzik et al. [31] reported that the switch from neonatal exon 6a to the adult exon 6 variant results in improved cardiac excitability. In the ‘adult’ Nav1.5, peak current densities increased, and steady-state activation shifted toward hyperpolarized potentials. The shift toward hyperpolarized potentials may be associated with an increased window current, which contributes to QTc prolongation. The authors also speculated that SCN5A mutations associated with sudden infant death syndrome may result in severe channel defects when the neonatal exon 6 is replaced with the adult variant after birth. The findings of this study also suggested that the replacement of the ‘neonatal’ Nav1.5 with the ‘adult’ Nav1.5 improved cardiac excitability. Additionally, the R1623Q mutation in the ‘adult’ Nav1.5 increased the probability of channel opening. Thus, we speculated that ‘neonatal’ Nav1.5, which is expressed during the prenatal period, suppresses the opening probability of R1623Q and ameliorates QTc prolongation. However, as the ‘adult’ Nav1.5 is replaced around the late-term and neonatal period, severity of R1623Q mutation might vary by those expression levels. Earlier onset severe congenital LQT3 caused by certain *SCN5A* mutations, including R1623Q, may be inversely correlated with the rate of expression of ‘neonatal’ Nav1.5. Developmental switching of neonatal/adult Nav1.5 isoforms might explain part of the mechanisms underlying severe LQT syndrome in fetuses and neonates.

### Study Limitations

The patient cells used in this study exhibited a heterogenous *SCN5A* genotype comprising both the R1623Q mutant gene and the unmutated reference gene (WT). However, homogenous transfectants with only the mutated gene or the control reference gene were analyzed in this study. Therefore, cells that have heterogenous *SCN5A* genotype must be examined in the future. The mutated gene may exert a dominant-negative effect.

We did not perform automated patch clamp recording at physiological temperature because of the instability of cells at that temperature. This might lead to an underestimation of the late I_Na_ of Nav1.5 R1623Q [41].

The Nav1.5 (α) channel interacts with β1–4 channels to generate currents. Therefore, the effect of the interaction between β channels and α channels must be analyzed in the future, as β channels exhibit differential expression in adults and neonates/fetuses.

The expression ratio of ‘neonatal’ and ‘adult’ Nav1.5 variants in fetuses, infants, and adults has not been precisely elucidated. Further studies are needed to confirm role of ‘neonatal’ in the LQT syndrome in fetuses and neonates.

## 5. Conclusions

In this study, an iPSC cardiomyocyte model was established from a patient with LQTS harboring a heterozygous Nav1.5 R1623Q mutation. The patient iPSC cardiomyocyte model exhibited a prolonged FPDcF (analogous to QTcF) compared to that of healthy volunteers. FPDcF was shortened with mexiletine treatment, and the frequency of arrhythmia-like EAD events following E4031 administration increased. These characteristics partly reflect the patient phenotypes. Since the R1623Q mutation is related to severe congenital LQT syndrome in fetuses and neonates, the effect of neonatal variants on the electrophysiological properties of the R1623Q mutant was examined using an automated patch-clamp system. The steady-state activation of the ‘neonatal-R1623Q’ Nav1.5 shifted towards depolarized potentials, suggesting a loss-of-function effect by the neonatal isoform. Both R1623Q and neonatal R1623Q delayed inactivation of I_Na_ and increased the late Na current. However, we did not observe synergistic effects by the ‘neonatal’ and ‘R1623Q’ on inactivation τ and late Na currents, which prolonged APD and led to reentrant ventricular tachycardia and TdP. We speculated that the neonatal Nav1.5 isoform may ameliorate the QTc prolongation. The developmental switching of the neonatal/adult Nav1.5 isoforms may play a role in the mechanisms underlying severe LQT syndrome in the fetuses and neonates, which should be elucidated in the future.

## Figures and Tables

**Figure 1 biology-10-01062-f001:**
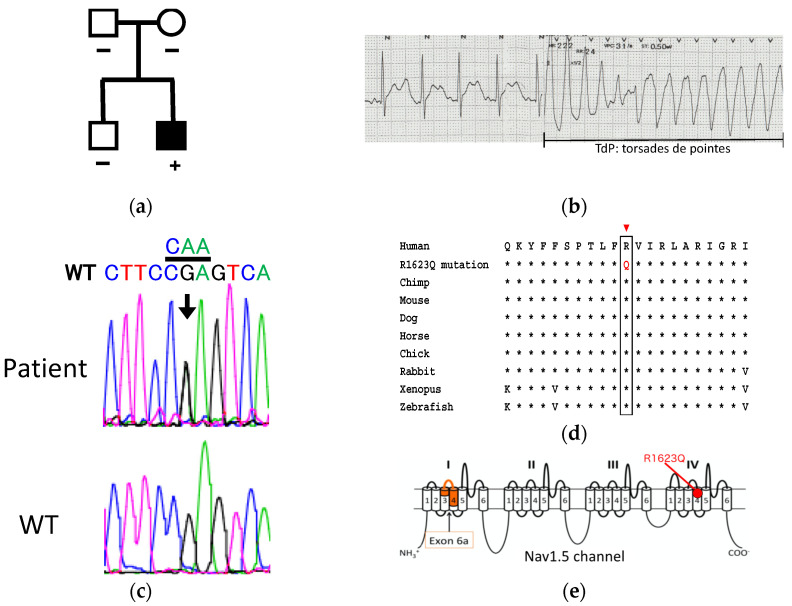
Clinical summary of the patient harboring Nav1.5 R1623Q mutation. (**a**) Pedigree of the family. (**b**) Baseline ECG of the index patient when he had VT storm at age of 2 years. QTc 540 ms was diagnosed at this time. (**c**) Sequence analysis of *SCN5A* identified the R1623Q heterozygous point mutation in the patient genome. (**d**) Multiple amino acid-sequence alignments of Nav1.5 regions bearing the R1623Q mutation with corresponding Nav1.5 amino acid sequences of different species. (**e**) Membrane-spanning model of the Nav1.5 channel indicating the locations of exon 6a and R1623Q mutation.

**Figure 2 biology-10-01062-f002:**
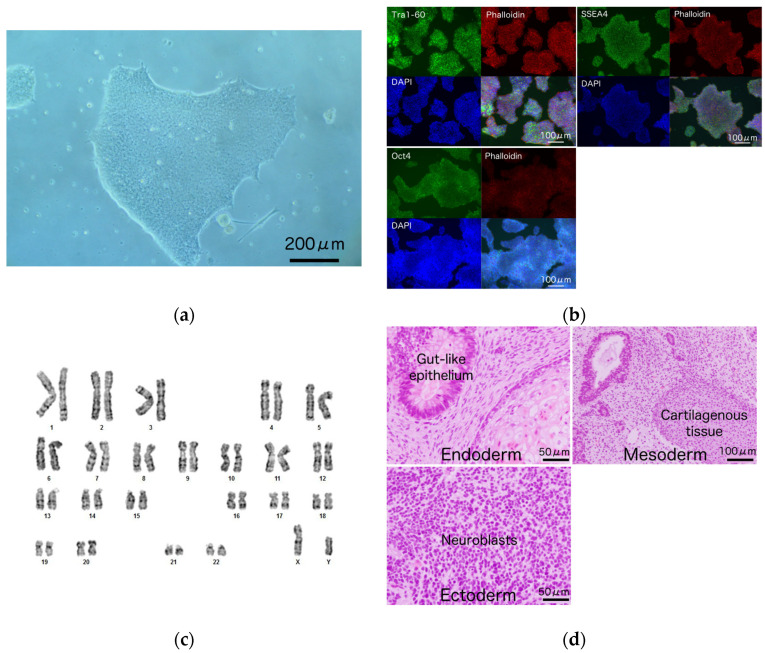
Characterization of induced pluripotent stem cell (iPSC) from a patient with Nav1.5 R1623Q mutation. (**a**) iPSC colonies derived from the lymphoblastoid cell lines (LCLs) of a patient with Nav1.5 R1623Q mutation. Scale bar = 200 μm. (**b**) The expression of the pluripotency markers TRA-1-60, SSEA4, and OCT4 in R1623Q mutation-harboring iPSC colonies was analyzed using immunostaining. Scale bars = 100 μm. (**c**) Karyotype of the iPSCs harboring R1623Q mutation was normal. (**d**) Teratomas were prepared from iPSCs harboring the R1623Q mutation. The tissues from all three germ layers were detected in the teratoma. Scale bars = 50 μm for ‘endoderm’ and ‘ectoderm’; 100 μm for ‘mesoderm’.

**Figure 3 biology-10-01062-f003:**
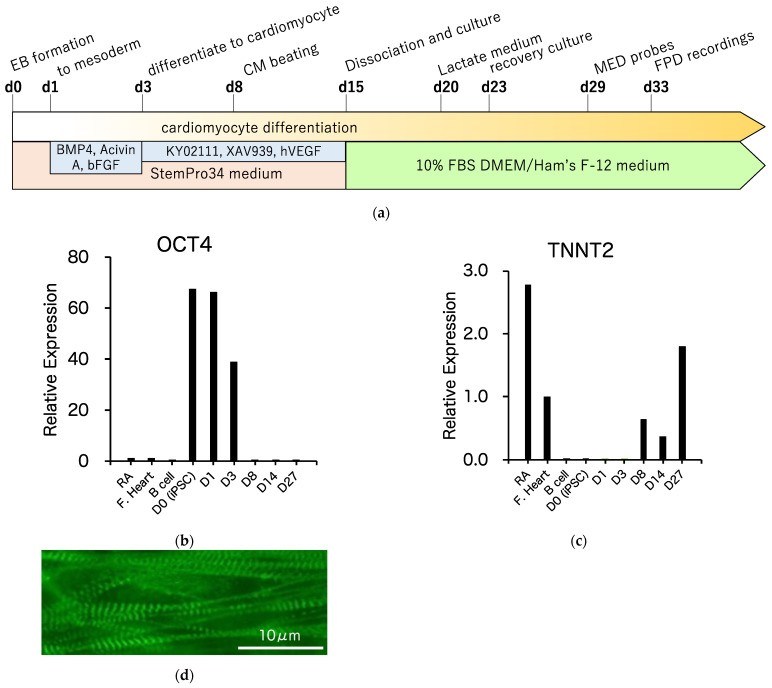
Cardiomyocyte differentiation from induced pluripotent stem cells (iPSCs). (**a**) Schedule of cardiomyocytes differentiation from iPSCs. EB: embryoid body, BMP4: bone morphogenetic protein 4, hVEGF: human vascular endothelial growth factor. Expression of pluripotency-associated (**b**) and cardiac markers (**c**) in differentiated cardiomyocytes. RA: adult right atrium, F. Heart: fetal total heart. (**d**) Immunostaining analysis revealed the presence of the contractile protein α-actinin in the cardiomyocyte derived from iPSCs. Scale bar = 10 μm.

**Figure 4 biology-10-01062-f004:**
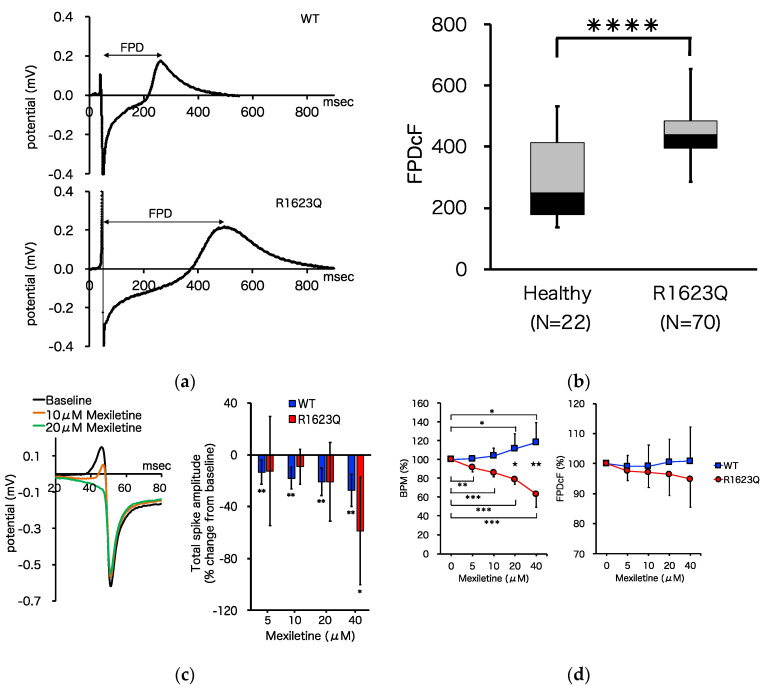
Electrophysiological analysis of induced pluripotent stem cell (iPSC) derived cardiomyocytes (CMs). (**a**) Representative baseline field potential waveforms in iPSC-CMs from healthy volunteers (upper) and patients with R1623Q mutation (lower). (**b**) The FPDcF of WT iPSC-CMs was significantly different from that of R1623Q mutation-harboring iPSC-CMs (WT, *n* = 22; R1623Q, *n* = 70, **** *p* < 0.0001). (**c**) Representative traces showing the Na^+^ spike overlaid at the baseline and increasing concentration of mexiletine. The bar graph presents the mean ± standard deviation (SD) percentage change in total spike amplitude at increasing mexiletine concentrations (WT, *n* = 7; R1623Q, *n* = 6). * *p* < 0.05, ** *p* < 0.01, baseline vs. each dose. There were no significant differences in percent changes between WT and R1623Q at each dose level. Dose–response plots showing the effects of mexiletine (WT, *n* = 10; R1623Q, *n* = 7) (**d**) and E4031 (WT, *n* = 33; R1623Q, *n* = 28) (**e**) on BPM and FPDcF. All data were normalized to the baseline value. Values are represented as mean ± SD. * *p* < 0.05, ** *p* < 0.01, *** *p* < 0.001, Baseline vs. each dose. ★ *p* < 0.05, ★★ *p* < 0.01 WT vs. R1623Q. (**f**) Screenshots of representative raw tracings showing early afterdepolarization (EAD) and triggered activity (TA). (**g**) The incidence of EAD events after E4031 administration in MED analysis (WT, *n* = 34; R1623Q, *n* = 38).

**Figure 5 biology-10-01062-f005:**
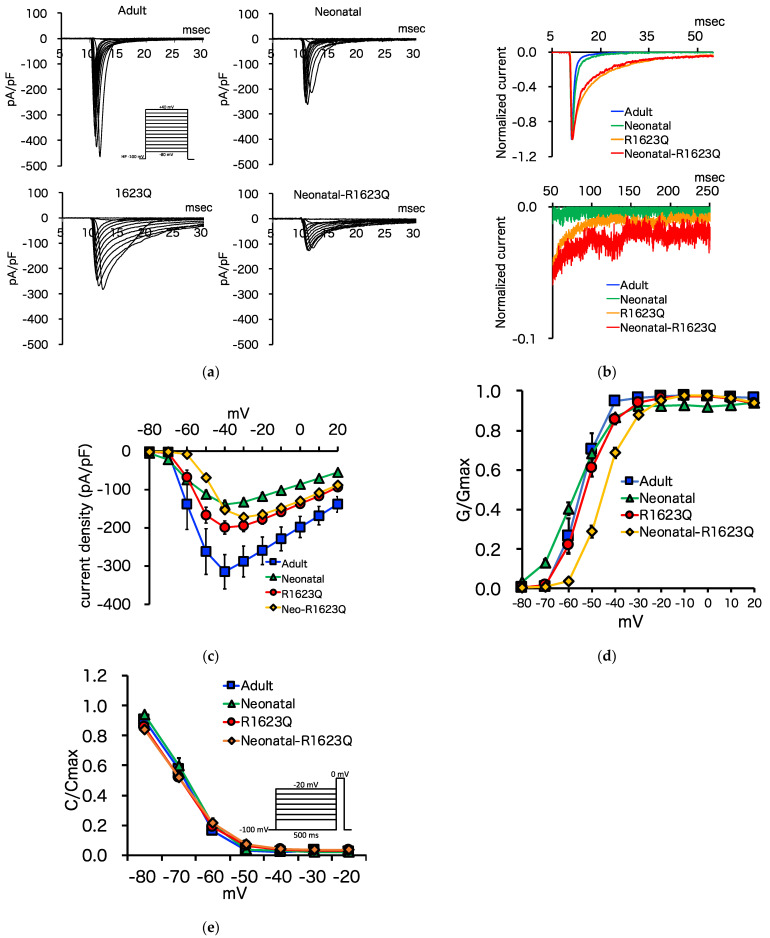
Whole-cell current recordings of ‘adult,’ ‘neonatal,’ ‘R1623Q,’ and ‘neonatal-R1623Q’ mutant Nav1.5 channels using automated patch-clamp. (**a**) Whole-cell I_Na_ was elicited with the pulse protocol shown in the inset. (**b**) Normalized I_Na_ for ‘adult’ (blue), ‘neonatal’ (green), ‘R1623Q’ (red), and ‘neonatal-R1623Q’ (orange). (**c**) Current–voltage relationship of peak I_Na_ for ‘adult’ (blue), ‘neonatal’ (green), ‘R1623Q’ (red), and ‘neonatal-R1623Q’ (orange) variants was obtained by normalizing the peak current amplitude from pulse protocols shown in (**a**) to the cell capacitance. (**d**) The voltage dependence of channel activation (G-V) curves was represented for ‘adult’ (blue), ‘neonatal’ (green), ‘R1623Q’ (red), and ‘neonatal-R1623Q’ (orange) Nav1.5. (**e**) The voltage dependence of channel inactivation of ‘adult’ (blue), ‘neonatal’ (green), ‘R1623Q’ (red), and ‘neonatal-R1623Q’ (orange) Nav1.5. (**f**) Experimental data obtained for the current–voltage relationship were used to determine the inactivation time constant in the range of potential from −40 to 40 mV. Current decay after the peak I_Na_ was fitted to a bi-exponential function. The resulting time constant (τ) values were plotted against the applied voltage for ‘adult’ (blue) and ‘R1623Q’ (red) Nav1.5 (**left** graph), and for ‘neonatal’ (green) and ‘neonatal-R1623Q’ (orange) Nav1.5 (**right** graph). Two-way ANOVA test detected the significant difference (* *p* < 0.05) in the variables (WT vs. R1623Q mutation); further, the Sheffé test was performed. The statistical analysis results are summarized in the table below the graphs. In the table, ‘a’ denotes the comparison between ‘adult’ and ‘neonatal’, ‘b’ denotes the comparison between ‘adult’ and ‘R1623Q’, ‘c’ denotes the comparison between ‘adult’ and ‘neonatal-R1623Q’, ‘d’ denotes the comparison between ‘neonatal’ and ‘R1623Q’, ‘e’ denotes the comparison between ‘neonatal’ and ‘neonatal-R1623Q’, and ‘f’ denotes the comparison between ‘R1623Q’ and ‘neonatal-R1623Q.’ Asterisks represent the significant differences between variant channels: *p* < 0.05. (**g**) Late I_Na_ currents increased in both ‘R1623Q’ and ‘neonatal-R1623Q’ Nav1.5. Two-way ANOVA test detected the significant difference (* *p* < 0.05) in the variables (WT vs. R1623Q mutation) only in the 50 msec data. The Sheffé test was performed for the 50 msec data and the resulting significance (* *p* < 0.05) is indicated in the corresponding graph.

**Table 1 biology-10-01062-t001:** Summary of electrophysiological data.

Parameter	‘Adult’ Nav1.5	‘Neonatal’ Nav1.5	‘R1623Q’ Nav1.5	‘Neonatal-R1623Q’ Nav1.5	Two-Way ANOVA (*p*-Value)		Interaction (*p*-Value)
					Adult vs. Neonatal	WT vs. R1623Q	
Peak sodium current density (pA/pF)	−315.0 ± 44.9	−139.5 ± 11.0	−199.3 ± 17.3	−172.6 ± 9.9	*	*	* a, * b, * c
Steady-state activation (V_1/2_, mV)	55.4 ± 1.3	−57.9 ± 1.0	−52.6 ± 1.2	−45.0 ± 0.8	*	*	* c, * d, * e, * f
Steady-state activation (k)	4.4 ± 0.4	4.9 ± 0.3	4.7 ± 0.3	4.6 ± 0.2			-
Steady-state inactivation (V_1/2_, mV)	−70.1 ± 1.0	−68.0 ± 1.1	−70.7 ± 0.8	−70.3 ± 0.6			-
Steady-state inactivation (k)	4.4 ± 0.1	4.8 ± 0.1	6.0 ± 0.1	6.8 ± 0.2	*	*	* b, * c, * d, * e, * f

Values are expressed as mean ± standard error of mean. ‘a’ denotes comparison between ‘adult’ and ‘neonatal’, ‘b’ denotes comparison between ‘adult’ and ‘R1623Q’, ‘c’ denotes comparison between ‘adult’ and ‘neonatal-R1623Q’, ‘d’ denotes comparison between ‘neonatal’ and ‘R1623Q’, ‘e’ denotes comparison between ‘neonatal’ and ‘neonatal-R1623Q’, ‘f’ denotes comparison between ‘R1623Q’ and ‘neonatal-R1623Q.’ Asterisks (*) represent significant differences between variant channels: *p* < 0.05.

## Data Availability

Not applicable.

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
