# Peer review of "Induced Pluripotent Stem Cell-Derived Cardiomyocytes with SCN5A R1623Q Mutation Associated with Severe Long QT Syndrome in Fetuses and Neonates Recapitulates Pathophysiological Phenotypes"

_biology, 2021, doi:10.3390/biology10101062_

Round 1

Reviewer 1 Report

In this study, Dr. Hayama and colleagues investigated the electrophysiological consequences of R1623Q Nav1.5 mutation commonly observed in the clinic as LQTS type-3 in fetuses/infants and employed both iPSC-CMs and HEK293 expression line to unravel the pathophysiology of the disease. Overall, the experiments were done nicely, the methods were explained in details and the findings are quite extensive. The availability of the sequencing data and the patient data also provided a solid background information for this study. I personally enjoyed reading the manuscript as well, although some questions and comments need to be addressed by the authors:

  • One of the main issues is that the experiments were done in room temperature. Although conducting sodium channel experiments in physiological temperature is known to be challenging, INa may exhibit temperature-dependent effect which affects the electrophysiological properties of the mutant and also modulates the effects of pharmacological compounds (https://www.nature.com/articles/s41598-018-22033-1). I would strongly suggest the authors to conduct similar experiments in physiological temperature to confirm their findings. 
  • In section 3.1, the authors need to mention the sex of the proband. Although it was clear from the pedigree that the patient is male, it is also necessary to mention in the text.
  • The presentation and quality of the figures need be improved. Some are too small and not visible, and the arrangement is very messy. The authors could arrange these figures in other softwares, for example powerpoint or adobe illustrator, and then bring them back to the manuscript in a good shape. Please make sure that the font size is not too small.
  • Line 414: "FPDcF (by up to 150% and 110% in WT and R1623Q mutant-harboring iPSC-CMs, respectively, from the baseline". This was not correctly reflected in Figure 4D. It is clearly seen that in the R1623Q, the FPDcF after 100 nM E4031 is less than the baseline. Please clarify and revise. 
  • All figures lack of statistical analyses. At present it is not clear whether the changes we saw are statistically significant or not. For example in Figure 4C-D and others.
  • Figure 4D: Please explain (or speculate) on the cause of FPD reduction following 100 nM E4031 in the mutant cells. I know that the authors have explained briefly in the discussion about the possible cause of non-changing FPD as compared to WT but they haven't explained about this reduction. It could be that the reduction is not statistically significant so please perform statistical analysis as well.
  • Section 3.5: I think the rationale of using HEK293 cells instead of iPSC-CM to study the Nav1.5 kinetics and current is still missing. The authors need to explain why they chose expression line than iPSC to be used in automated patch clamp. Several groups have demonstrated the applicability of Synchropatch to study iPSC-CM as well (https://www.biorxiv.org/content/10.1101/2020.05.08.084350v1.full) so it should be doable.
  • Line 485: The authors need to also describe the findings under ISO in the results section. These results are important and consistent with previous study which showed that ISO may abbreviate APD in already prolonged AP (https://www.frontiersin.org/articles/10.3389/fphys.2020.587709/full).
  • Line 500: The authors need to explain or speculate why the R1623Q mutation has more EAD like events than the WT despite the non-changing / reduced FPD under E4031? If FPD correlates well with APD, the non-existence of APD prolongation would not increase the occurence of EAD? 
  • Line 501: "prolonged action potential increase calcium influx through L type calcium channels and cause EAD", this is not entirely correct. The main cause of EAD is not a larger Ca2+ influx but because of ICaL reactivation and a larger ICaL window. Please read and cite this paper (https://www.frontiersin.org/articles/10.3389/fphys.2020.587709/full) as it explained the mechanisms of EAD generation.
  • I think it is also important to mention in the discussion that an increased late INa has been shown to be associated with EADs in iPSC-CM experiments and such phenomenon can induce reentry in the tissue levels.
  • Line 27: It should be "...E4031, an IKr blocker,..."
  • Line 62: the "+" in Na should be superscripted ("Na+"). Also, please check the writing of other "Na" throughout the manuscript.
  • Line 400: should be "MED64" without a space in between.
  • Line 410 and 414: maybe "elongation" should be "prolongation"? They are not interchangeable and have different meaning. Please clarify which is more appropriate.
  • Figure 4F: please add the y-axis label.
  • Line 425: please confirm whether "rates" or "incidence" is more appropriate.
  • Line 431: It could be useful to specify that the 293T cells are "HEK 293" cells.
  • Line 436: "...‘neonatal-R1623Q’ Nav1.5 were lower than those of the ‘adult’ variant." Please specify, which part of the traces were lower? The amplitude of the peak? The tau? or something else?
  • Line 501: There is no "Figure 4G" in this manuscript. Please clarify.
  • Line 537: should be "individual"

Reviewer 2 Report

The authors explore the impact of a LQT3 SCN5A variant (R1623Q) in the ‘adult’ and ‘neonatal’ Nav1.5 splice variant using heterologous 293T cell expression system. Subsequently, the authors created iPSC derived cardiomyocytes (iPSC-CM) to evaluate the impact of the variant in an in vitro cell model that better recapitulates the in vivo situation, as these cells express a more complete repertoire of cardiac myocytes proteins (including auxiliary subunits of the Nav1.5 channel). Although their iPSC-CM clonal line recapitulates the pathophysiological condition, the authors do not fully exclude that the observed difference with the control line is by the R1623Q variant.

Major comments:

1) The iPSC-CM model suffers from inter-clonal differences and even differences are observed between differentiation batches of the same line. In their study the authors use only one line, if multiple different lines were generated the authors should report their clonal variability. Besides evaluating different clonal lines, the authors should also mention the batch-to-batch variability, i.e. how many times were the clonal iPSC lines differentiated into cardiac myocytes.

2) The ideal control to evaluate the impact of the R1623Q variant in the iPSC-CM model is to create an isogenic control.

3) Since iPSC-CMs are immature compared to native cardiac myocytes, the cells express also the neonatal Nav1.5 isoform. To correlate the data of the heterologous 293T cells, the amount of ‘adult’ and neonatal’ variant in their iPSC-CM can be determined.

4) The authors evaluate the sodium and hERG current expression in their FP (MEA) experiments pharmacologically using the drugs mexiletine and E4031. Although this is an elegant approach, the absence of a response to the sodium channel dug mexiletine can also be because the sodium channels are all inactivated and not available (for example by a depolarized membrane potential). The authors can evaluate the presence of the sodium current by analyzing the initial upstroke of their FP recordings (slope of the peak). If sufficient sodium current is expressed, the slope should be steep.

5) To correlate the data from the heterologous ‘adult’ and 'neonatal’ Nav1.5 WT and R1623Q variant expression in 293T cells with that of their iPSC-CM, the sodium current in the iPSC-CM should be characterized by patch-clamp.

6) In their analysis of the ‘adult’ and ‘neonatal’ Nav1.5 WT and R1623Q variant expression in 293T cells, the authors created stable expressing cell lines. The authors compare the peak current expression between all lines but this difference can be the consequence of the cell line itself (for example the integration site of the plasmid) and does not reflect on possible expression differences due to the ‘adult’ and ‘neonatal’ variant or the presence of the R1623Q variant. Emphasis should be on the difference in the kinetics and it appears that the ‘neonatal’ R1623Q variant displays a more depolarized threshold of channel activation, and therefore a loss-of-function, compared to the ‘adult’ variant. This loss of function of the ‘neonatal’ R1623Q variant in heterologous expression system does not correlate with the more severe pathophysiology in fetuses and neonates. The authors might elaborate on this observation in their discussion.

Minor comment

7) The figures should be optimized, in several figure panels the labels on the axes are to small and hardly readable.

Round 2

Reviewer 1 Report

Thank you for the clear response to my previous comments. I do have some minor remarks to be addressed:

  • Line 416: The initial upstroke in Figure 4c is not visible, therefore the effect of mexiletine can't be observed. Please correct the figure visualization (y axis).
  • Abstract: "Ikr" should be "IKr"

Reviewer 2 Report

The authors addressed most of my concerns as detailed in their response. However, part of this information should be included in the manuscript or provided as supplementary information. Especially the information on the number of iPSC lines created and state that a single clone was selected for the represented data and that it is not pooled data. It is advised that the authors also mention the selection criteria used for selecting the best iPSC line and details on inter-clonal and inter-batch variability are always welcome.

The authors addressed the suggestion to analyse the slope of their field potential recordings to investigate the presence of Na current expression. This is now represented in figure 4C. However, on this scale the steepness of the slope is not visible and the authors should also represent their obtained  values before and after drug administration in both control and patient derived iPSC-CM. A good representation of this parameter and the analyzed values is mandatory as, in the absence of patch-clamp data, this value is the main indication that the sodium current is contributing to the recorded field potentials, which reflect on the cell's action potential.

The quality of some figures can still be improved. In some figures the font size of the scale bar labels differ substantially between panels. 
